# A Comparative Study of Ethanol and Citric Acid Solutions for Extracting Betalains and Total Phenolic Content from Freeze-Dried Beetroot Powder

**DOI:** 10.3390/molecules28176405

**Published:** 2023-09-02

**Authors:** Rahul Kumar, Lisa Methven, Maria Jose Oruna-Concha

**Affiliations:** Department of Food and Nutritional Sciences, University of Reading, Whiteknights, Reading RG6 6DZ, UK; r.kumar@pgr.reading.ac.uk (R.K.); l.methven@reading.ac.uk (L.M.)

**Keywords:** extraction, betalain pigments, total phenolic content, ultrasound, optimisation, response surface methodology, artificial neural network

## Abstract

This research compares the extraction of betalains (betacyanin and betaxanthin) and total phenolic content using citric acid and aqueous–ethanol solutions. The aim is to find an environmentally sustainable alternative solvent for extracting these compounds from dried beetroot powder. Using citric acid solution as a solvent offers several benefits over ethanol. Citric acid is a weak organic acid found naturally in citrus fruits, making it a safe and environmentally friendly choice for certain extraction processes. Moreover, the use of citric acid as solvent offers biodegradability, non-toxicity, non-flammability, and is cost effective. A full factorial design and response surface methodology (RSM) were employed to assess the effects of extraction parameters (extraction time (5–30 min), extraction temperature (20, 30, 40 °C), pH of citric acid solution (3, 4, 5) and ethanol concentration (10, 20, 30% *v*/*v*)). The yield was determined spectrophotometrically and expressed as mg/g of dry powder. The results showed that citric acid solution yielded 85–90% of the ethanolic extract under identical conditions. The maximum yields of betacyanin, betaxanthin, and total phenolic content in citric acid solution were 3.98 ± 0.21 mg/g dry powder, 3.64 ± 0.26 mg/g dry powder, and 8.28 ± 0.34 mg/g dry powder, respectively, while aqueous–ethanol yielded 4.38 ± 0.17 mg/g dry powder, 3.95 ± 0.22 mg/g dry powder, and 8.45 ± 0.45 mg/g dry powder. Optimisation resulted in maximum extraction yields of 90% for betalains and 85% for total phenolic content. The study demonstrates the potential of citric acid as a viable alternative to polar organic solvents for extracting phytochemicals from plant material, providing comparable results to aqueous–ethanol. Artificial Neural Network (ANN) models outperformed RSM in predicting extraction yields. Overall, this research highlights the importance of exploring bio-solvents to enhance the environmental sustainability of phytochemical extraction.

## 1. Introduction

Beetroot (*Beta vulgaris* L.) is a herbaceous blooming biennial plant native to Asia and Europe that belongs to the Chenopodiaceae family [1] and can be grown across the seasons [2]. It is widely consumed as a salad, as a juice, or after pickling. It is known to contain high levels of nutritional and bioactive compounds including nitrates, phenolics, ascorbic acid, and water soluble pigments called betalains [3,4,5]. Beetroot and its juice consumption have been clinically proven to provide protection against non-communicable diseases (NCDs). The claimed health benefits of beetroot include functioning as an antioxidant, anti-depressant, anti-microbial, anti-fungal, anti-inflammatory, diuretic, expectorant, and in preventing liver and cardiovascular damage [6].

Industrial scale production, processing, packaging, retail market and household consumption of beetroot leads to a wastage of more than 50% across the United Kingdom (UK) [7]. The valorisation of beetroot and its waste can be achieved by extracting the natural pigment betalains and total phenolic compounds [8]. Betalains are classified into two different classes namely betacyanins (BC) and betaxanthins (BX) (Figure 1). These two nitrogenous compounds can be of significant importance to food, pharmaceuticals, cosmetics and dye industries, where it is also known as “beetroot red” [9]. The use of beetroot extracts as natural colourants in the food industry offers a variety of benefits. Fruit juices can be enhanced with a visually appealing colour, while bakery products like cakes and pastries can achieve vibrant reddish or pinkish hues in the dough or frosting. Yogurts and dairy products can achieve a pink or red tint naturally, and pasta can be made in colourful variations such as red or purple. Beetroot extracts are also incorporated into salad dressings and sauces to enhance their appearance, and they are used in confectionery products like candies and gummies to achieve red or purple colours without artificial additives. Furthermore, in the context of plant-based and vegan food products, beetroot extracts serve as an alternative to synthetic dyes for adding colour [10,11]. The stability of this colourant is pH- and temperature-dependent, and its application in high temperature processed products is limited. Betalains are stable in the pH range of 4–5 [12], with betacyanin remaining unchanged for at least 20 days at 4 °C and over 275 days when frozen at −30 °C. However, betalains are sensitive to heat and start degrading at temperatures above 50 °C. Boiling betanin-containing material causes a gradual colour change from red to yellowish-brown at 100 °C, leading to a decrease in both betacyanins and betaxanthins content. Higher temperatures and longer heating times result in more significant degradation of betalains. Overall, temperature and pH play crucial roles in determining the stability of betalains during storage and food processing [9]. However, its ready availability and low price has driven large-scale applications in the food industry [3].

Betalain pigments are mostly extracted from the whole tuber rather than just from the peels through various methods including the use of aqueous–ethanol, supercritical fluids and other organic solvents [13,14]. More recently, the extraction process has been intensified by using pulsed electric field, ultrasound and microwave technology, in order to avoid higher consumption of solvents, shorten the extraction time, and lowering of the extraction temperature [15]. The use of organic solvents has been heavily questioned in recent years due to the environmental impact of the solvents, such as volatile organic compound (VOC) emissions, hazardous waste generation, and non-renewable resource depletion as well as the safety concerns associated with their handling. Additionally, traditional extraction methods are energy-intensive. Hence, there is a growing interest in the development of extraction procedures using alternative solvents which are perceived to be greener, cleaner, safer, and easier to adopt [16], and the citric acid solution meets all of the aforementioned requirements to be a greener and cleaner solvent because it is easy to obtain, eliminate, and is safer and easier to handle [14,17,18]. In this context, citric acid solutions have been added to aqueous–ethanol solutions to extract betalains and total phenolic compounds. Lazăr et al. (2021) [17] have used aqueous–ethanol acidified with citric acid; however, they did not control the pH of the solvent mixture, and the stability and extraction yield of betalains are known to be highly dependent on pH [19]. On the other hand, Singh et al. (2017) [14] extracted betalains employing a similar mixture in the pH range 4–6 without varying the ethanol concentration. While these earlier publications show the addition of citric acid to be promising, it is difficult to ascertain the individual contribution of the two components, citric acid and ethanol, in the mixture. The present study aims to overcome the limitations of earlier studies by investigating extraction in citric acid solution and aqueous–ethanol solutions separately, and comparing the extracts obtained under otherwise identical conditions. This approach offers valuable insights into their solvent selectivity, yield of extraction, environmental impact, and process optimisation. This knowledge is essential for advancing sustainable extraction practices and enhancing the utilisation of betalains in various industries. The use of ultrasound to intensify extraction has also been explored. Ultrasound-assisted extraction (UAE) is an innovative and environmentally friendly extraction technique that has gained significant attention in recent years. This non-invasive method utilises high-frequency sound waves to enhance the extraction process, making it more efficient and effective compared to traditional extraction methods. UAE offers several advantages, including reduced extraction times, lower energy consumption, and decreased reliance on organic solvents, making it a greener alternative. The ultrasound waves create cavitation bubbles in the solvent, causing rapid changes in pressure and temperature, which facilitate the release of bioactive compounds from the source material. This technology has been successfully applied to extract various bioactive compounds, such as carotenoids, betalains, and polyphenols, from different plant and food matrices [20]. The independent and interactive effects of operating parameters such as strength of the ethanol solution and pH of citric acid solution, extraction time, extraction temperature and ultrasound application will be evaluated using response surface methodology (RSM) as well as artificial neural networks (ANN). ANN architecture was developed on the basis of optimised number of hidden neurons, least mean square error (MSE), least root mean square error (RMSE), and highest coefficient of determination (R^2^). These statistical and model parameters played an extensive role in the ANN design completing the aim of the work.

## 2. Results

A full factorial design was implemented for the aqueous–ethanol extraction as well as extraction with citric acid solution, as shown in Table 1. Since betalains are readily water-soluble, extraction was also carried out using pure water as the solvent, however, the yields were significantly (*p* < 0.05) lower than with aqueous–ethanol and citric acid solution (data not reported). The stability of betalains is also likely to be reduced at a higher pH of water. Moreover, during preliminary studies, extraction was performed with aqueous–ethanol and citric acid solution without the application of ultrasound and the yield was significantly lower (*p* < 0.05) (data not reported), as similarly reported before (Nutter et al., 2021 [21]). Hence, data are only reported for aqueous–ethanol and citric acid solutions with ultrasonic extraction.

### 2.1. Effect of Extraction Time on Betalains and Total Phenolic Content

Figure 2 shows the effect of time, temperature and pH on the extraction of betalains and total phenolic content in citric acid solutions, whereas Figure 3 shows data for extraction using aqueous–ethanol solutions as the solvent.

Extraction time was not a significant factor for extraction of betalains and total phenolic content when citric acid was used as the solvent (Table 2), and this is well illustrated by the response surface plots in Figure 2. This conclusion applies to the time range covered in this work (5–30 min), which is expected given the ready solubility of the betalains in citric acid solution and cavitational effect of ultrasound to perform a quick and effective extraction [22]. Short extraction times will only have an effect if there are mass transfer limitations [17,21]. Long extraction times, such as over 50 min, can result in lower yields due to degradation of the extract caused by free radical formation in the presence of ultrasound [17,21]. Using citric acid solutions as a solvent, the maximum levels extracted per g of dried beetroot powder were 3.98 mg BC, 3.64 mg BX and 8.28 mg GA/g as a measure of TPC. There were no significant interactions of time with temperature, nor of pH with time on the extraction of betalains and TPC. Therefore, it can be concluded that for the extraction of betalains and TPC from dried beetroot using citric acid solution as the solvent, time can be kept to a minimum in order to optimise the process. 

However, when aqueous–ethanol was used as solvent, the extraction time was significant for BC and TPC, but not for BX (Table 2). Figure 3 illustrates that betalain content increased with time, which could be attributed to the cavitational and thermal effects of ultrasound treatment [21,22,23,24]. Using aqueous–ethanol as a solvent, the maximum amounts extracted per g of dried beetroot powder were 4.38 mg BC, 3.95 mg BX and 8.45 mg GA/g as a measure of TPC. The yield obtained in this study was comparable or better than previous findings [17,21,23,25]. The interactive effect of time and ethanol concentration had significant positive effect on extraction of BC and BX at lower temperature range and this could be easily depicted from Figure 3, which was dominated by time. It can be concluded that time had a positive effect on the extraction using aqueous–ethanol as solvent and the process should be optimised for optimum time to enable maximum yield of betalains and TPC. In addition, when comparing the extraction efficiency for both solvents, the percentage yield using citric acid as a solvent in comparison to aqueous was over 91% for BC, 92% for BX and 98% for TPC, respectively.

### 2.2. Effect of Extraction Temperature on Betalains and Total Phenolic Content

Temperature had a significant effect on the extraction of BC, BX and TPC using either solvent (citric acid or aqueous–ethanol) (Table 2). Using either solvent, increasing the temperature from 30 to 40 °C tended to decrease the amount of betalains (BC and BX) in the resulting extract (Figure 2 and Figure 3a–f). However, the effect of temperature on TPC varied between the solvents, as the effect of temperature was limited with citric acid but substantial with ethanol. When using aqueous–ethanol, increasing the temperature from 30 to 40 °C increased the yield of TPC substantially, especially at higher ethanol concentrations (20%, 30%) (Figure 3). 

The decrease in betalain (BC and BX) and TPC content in the citric acid extracts with increasing temperature is due to their heat sensitivity. It is evident from Figure 2a–i. This observation is consistent with previously published data [17,26,27]. Janiszewska-Turak et al. (2021) [26] observed first-order degradation kinetics for betalains in the temperature range between 60–90 °C. At higher temperature and low pH betalain content decreased due to the instability of BC and BX at higher temperature and lower pH, as expected from previous studies [28,29,30]. The negative effect of higher extraction temperature on phenols and betalains was in good agreement with previously reported studies [25,31].

Using aqueous–ethanol as the solvent, temperature again had a significant negative effect on recovery of BC (Figure 3a–c) and BX (Figure 3d–f), explained by the thermal degradation of betalains [32]. However, extraction temperature significantly (increased the TPC content of the extract (Figure 3g–i). This increased extraction could be attributed to the enhanced damage to cell membranes of the beetroot powder by temperature and greater permeability for the solvent, coupled with the greater thermal stability of phenolics compared to betalains [21,23,25]. 

### 2.3. Effect of Solvent Type on Extraction of Betalains and Total Phenolic Content

Table 2 shows that the pH of the citric acid solution was the most influential parameter for the extraction of betalains and TPC using citric acid, as previously reported [14,17]. Increasing pH significantly increased the yield of BC, BX and TPC (Table 2, and Figure 2a–i). The results of this study are in agreement with previous studies that the concentration of BC extracted from beetroot was higher than of BX [21,23,24,33], The results of this study are in agreement with previous studies that the concentration of BC extracted from beetroot was higher than of BX [5], which may be due to BC naturally occurring at higher levels than BX in beetroot, or due to the greater stability of BC than BX on extraction. 

Using aqueous–ethanol as the solvent, the ethanol concentration had a significant effect (Table 2) and was the second most influential parameter for the extraction of BC, BX, and TPC after temperature. The increase in ethanol concentration tended to reduce the extraction of BC and BX (Figure 2a–f), but it increased the extraction of TPC (Figure 3g–i). The negative effect of ethanol concentration on betalains was similar to the findings of previous trials [25] and may be due to the ability of ethanol to extract multiple components at a time at higher concentration. Roriz et al. (2017) [34] reported that an ethanol concentration above 20% compromised the extractability of betalains and they attributed this to the increased affinity of other ethanol soluble substances at the solvent levels. The combined effect of increasing ethanol concentration and time tended to reduce the concentration of betalains in the extract (Figure 3a–f), whereas no combined effect on the total phenolic content was observed. On the other hand, it was observed that the concentration of betalains in the extract increased with ethanol concentration until at temperature of 30 °C was reached and then decreased with increase in temperature (above 30 °C), which is well illustrated in Figure 3. This could be explained by the swelling of the cellular structure with an initial increase in temperature, whereas further increases led to the thermal degradation of betalains [25,33,34]. These findings illustrated that for process optimisation, the concentration of ethanol could be reduced and this could be environmentally and economically beneficial by reducing solvent consumption.

### 2.4. Modelling, Prediction and Optimisation by RSM

#### 2.4.1. Citric Acid Solution as an Extraction Solvent

The response models obtained from RSM analysis for BC, BX, and TPC is given below in Equations (1)–(3). The same models were also used for performing the prediction, optimisation of the extraction process, and comparing the prediction ability of RSM against ANN.
Betacyanin (BC) = −9.18 + (0.323 × Temperature) + (2.637 × pH) − (0.0342 × Temperature × pH) − (0.045 × Temperature^2^)(1)
Betaxanthin (BX) = −4.69537 + (0.063 × Temperature) + (2.793 × pH) − (0.043 × Time × Temperature) + (0.002 × Temperature × pH) − (0.018 × Temperature^2^)(2)
Total phenolic content (TPC) = −16.7401 + (0.536 × Temperature) + (7.073 × pH) − (0.078 × Temperature × pH) − (0.005 × Temperature^2^)(3)

Correlations between experimental and predicted results using RSM models given in Equations (1)–(3) can be seen in Figure 4. Coefficient of determination (R^2^) between predicted and experimental values were 0.78, 0.89, and 0.79 for BC, BX, and TPC, respectively, indicating a good fit of the model as previously reported [14,33,35]. Furthermore, RSME was also calculated to quantify the deviation of the predicted values from the experimental values. The RMSE values for BC, BX, and TPC was computed to be 0.60, 0.28, and 0.67, respectively. The RMSE values are relatively high, and it was evident from the scattered pattern of predicted and experimental data across the central prediction line in Figure 4a–c.

For performing optimisation, Equations (1)–(3) were used, which leads to the maximum yield of betalains and TPC with tested variable range. The resulting optimised conditions were 10 min of extraction time at 30 °C of extraction temperature and pH 5, with yields of BC, BX, and TPC of 3.95 mg of BC/g, 3.54 mg BX/g, and 7.17 mg of GA/g of dried beetroot powder, respectively, with an RSM desirability value of 0.928. The optimised condition and responses were validated via a real-time experiment in triplicate as shown in Table 3. The significance of the interaction between the variables evaluated in the experimental design was used to define this condition.

#### 2.4.2. Aqueous–Ethanol as an Extraction Solvent

The response models obtained from RSM analysis for BC, BX, and TPC are given below in Equations (4)–(6), respectively. The same models were also used for performing the prediction, optimisation of the extraction process, and comparing the prediction ability of RSM against ANN.
Betacyanin (BC) = 2.97675 + (0.029 × Time) + (0.096 × Temperature) − (0.043 × Ethanol) − (0.0711 × Time × Ethanol) + (0.089 × Temperature × Ethanol) − (0.019 × Temperature^2^)(4)
Betaxanthin (BX) = 2.20591 + (0.092 × Temperature) − (0.013 × Ethanol) − (0.007 × Time × Ethanol) − (0.002 × Temperature^2^)(5)
Total Phenols (TPC) = 9.1896 + (0.062 × Time) − (0.165 × Temperature) +(0.004 × Ethanol) + (0.004 × Temperature × Ethanol) − (0.002 × Time^2^) + (0.002 × Temperature^2^) − (0.003 × Ethanol^2^)(6)

Figure 4 shows the correlation between the predicted values using the RSM models given in Equations (4)–(6) and the experimental results. The values of co-efficient of determination for BC, BX and TPC was calculated to be 0.88, 0.79, and 0.86, respectively. The obtained values were greater than those reported before [14], but are in close agreement with other authors [25,35]. The RMSE for BC, BX, and TPC was calculated to be 0.21, 0.30, and 0.26, respectively. The lower values of RMSE are well illustrated by less scattering of data across the central prediction line as shown in Figure 4d–f, and this could be attributed to the low range of variation within the data [36].

For performing optimisation, Equations (4)–(6) were used. The resulting optimised conditions were 15.8 min of extraction time at 20.1 °C of extraction temperature and 10% of ethanol concentration in water, with yields of BC, BX, and TPC as 4.15 mg of BC/g, 3.52 mg of BX/g, and 7.71 mg of GA/g of beetroot powder, respectively, with an RSM desirability value of 0.679. The optimised condition and responses were validated via a real-time experiment in triplicate as shown in Table 3. The significance of the interaction between the variables evaluated in the experimental design was used to define this condition. 

### 2.5. ANN Modelling, Prediction and Comparison with RSM 

#### Predictive Model Development with ANN

The design of experiments of RSM with responses was adopted for developing an additional predictive model with ANN to compare with RSM. The total number of datasets for this ANN-based machine learning approach was equal to the number of experimental results shown in Table 4 and Table 5. To train the model partitioning of the data was important to avoid overfitting of the model and over parameterisation of the functions [37]. It was partitioned as 70%, 15% and 15% for training, testing, and validation, respectively [37,38,39]. Training was carried out for 1–60 neurons in series; for each increase in number of neurons, the predicted value was compared with experimental results and RMSE was calculated. The predicted yields by RSM and ANN are shown in Table 4 and Table 5. The selection of the optimised number of neurons in hidden layer was based on the least RMSE value. 

For citric acid solution as extraction media, the ANN-based prediction illustrated promising results in terms of RMSE, MSE and R^2^, and the predicted values using the ANN model had great accuracy, as evidenced by the scattering of the data across the central prediction line. The obtained values of RMSE, MSE and R^2^ are shown in Table 6 and Table 7. The total number of optimised neurons in the hidden layer for citric acid solution as extraction media was six with a lowest possible value of RMSE, which it was 0.0043, as shown in Figure 5a. If the number of neurons is very low or very high, it may cause underfitting or overfitting of the model [37]. The correlation between the predicted and experimental values for BC, BX, and TPC is shown in Figure 5b.

The predicted yields using the ANN models for aqueous–ethanol as extraction media are shown in Table 3. The obtained values of RMSE, MSE and R^2^ are shown in Table 6. The number of optimised neurons in the hidden layer for aqueous–ethanol as extraction media was eight, with a lowest possible value of RMSE, which was 0.0042, as shown in Figure 6a. The correlation between the predicted and experimental values for BC, BX, and TPC is shown in Figure 6b.

### 2.6. Prediction Performance Comparison for ANN with RSM

The prediction performance of ANN and RSM was compared in terms of their capacity to predict data as closely as possible to the original dataset. In terms of all statistical characteristics obtained from Table 6 and Table 7, it was discovered that the ANN tool was preferable. The R^2^ for ANN predicted data was found to be close to 0.99 for both type of the solvents and for their respective responses. On the other hand, the R^2^ for RSM predicted data for both type of the solvent varied between 0.78–0.89. R^2^ is not the only parameter to be checked, but it was the first check point for the comparison. Additionally, it was observed that ANN had 10-fold less error in terms of the MSE and RMSE compared to RSM. The value of RMSE and MSE for ANN ranged between 0.02–0.05 and 0.000049–0.0927, whereas for RSM, the range was 0.20–0.67 and 0.1011–0.2111, as illustrated in Table 6 and Table 7. The better accuracy of the former tool could be attributed to its universal ability to approximate non-linearity of the system, whereas RSM is restricted to a second-order polynomial [37,39,40]. Moreover, many studies reported that ANN modelling is a useful and flexible tool to generate models and to calculate the multiple responses in a single run [38]. This concludes that ANN could be a useful alternative predictive tool over conventional RSM.

### 2.7. HPLC Analysis

The individual betalains present in the optimised beetroot extract were identified against standards of betanin [41,42,43]. Figure 7A shows the HPLC elution profile at 538 nm of BC (13.85 min) and iso-betacyanin (IBC) (15.674 min) standards. Figure 7B,C shows the HPLC elution profiles for the optimised samples extracted using aqueous–ethanol and citric acid solution, respectively.

The most abundant betalain present in both the citric acid and aqueous–ethanol extracts was BC (3.89 ± 0.11 mg/g and 4.01 ± 0.08 mg/g of dried beetroot powder, respectively), followed by BX (3.42 ± 0.09 mg/g and 59 ± 0.09 mg/g of dried beetroot powder, respectively). 

## 3. Materials and Methods

### 3.1. Experimental Design

A full-factorial design was implemented for extraction using different concentrations of ethanol in water (10, 20, and 30% *v*/*v*) and citric acid solutions of variable pH (3, 4, 5) as solvents, coupled with ultrasonic parameters as shown in Table 1. The ethanol concentration range was in line with previous studies, where findings have reported that if too high the concentration of ethanol has a negative effect compound recovery; this was confirmed in our preliminary study (data not reported) and would also lead to higher solvent consumption [8,21,23,33,44]. The pH range of citric acid solutions was favourable for the stability of betalains [8,14,45]. Extraction time with ultrasound above 30 min was observed to have a negative effect in the preliminary study and in previous literature [21]. Hence, the time increased in 5 min interval up to 30 min. The extraction temperature was at three levels of 20, 30 and 40 °C, and not higher to minimise the risk of thermal degradation [8,25].

All experiments were carried out in triplicate. Means and standard deviation of the data were calculated for each treatment. Analysis of variance (ANOVA) was carried out to determine any significant differences (*p* < 0.05) between treatments and multiple pairwise comparisons were carried out using Tukey’s HSD test, using XLSTAT 2021.2 (Addinsoft, Paris, France).

### 3.2. Chemicals

Ethanol (purity > 99%), citric acid (purity > 95%), sodium hydroxide pellets, sodium phosphate dibasic (purity ≥ 99%), formic acid (Purity ≥ 98%), acetonitrile (LC/MS grade) and sodium carbonate were supplied by Fisher Scientific (Loughborough, UK). Folin–Ciocalteau reagent (2 M) was purchased from Scientific Laboratory Supplies Ltd. (Nottingham, UK). Standards of betanin (purity > 99%) and gallic acid (purity ≥ 98%) to measure betacyanin, betaxanthin and total phenolic content were purchased from Merck Chemicals Limited (Dorset, UK).

### 3.3. Sample Preparation

Fresh beetroot was purchased from a local supplier in Reading, UK. The beetroot was washed, cleaned, wiped, and chopped in a food processor (Kenwood Blend-X Fresh BLP41.A0GO, Kenwood Limited, Hampshire, UK). It was then transferred to an aluminium tray and subjected to blast freezing at −80 °C, for 24–36 h. It was subsequently freeze-dried (VirTis SP Scientific, Ipswich, UK) for 70–72 h until the moisture content dropped below 3% (dry weight basis). After freeze-drying, the samples were ground (Kenwood Prospero AT286 KW714229 Spice Mill, Kenwood Limited, Hampshire, UK) and sieved. The extraction experiments were performed using particles with an average diameter of 230 µm based on our previous research [46] as this particle size achieved the maximum extraction efficiency.

### 3.4. Extraction of Betalains 

#### 3.4.1. Ultrasound-Assisted Ethanolic Extraction 

Betalains were extracted according to the method described by Silva et al. (2018) [25] with some modifications. Freeze-dried beetroot powder (0.2 g, 230 µm) was extracted with 25 mL of aqueous–ethanol solvent (10%, 20% and 30% *v*/*v* ethanol), under continuous ultrasonication (Power 100 W; Frequency 42 kHz) at different time and temperature combinations as per the design given in Table 1. After the ultrasound treatment, the mixture was centrifuged (SIGMA, Laborzentrifugation 3K10, An der Unteren Söse, Germany) twice at 9384× *g* for 30 min to obtain a clear supernatant. The extract was then stored at 4 °C until analysis.

#### 3.4.2. Preparation of Citric Acid Solution

A 1 mM solution of citric acid, the solution was prepared using food grade crystalline citric acid. The pH of the obtained solution was 3.2–3.3, and further adjustment of the pH for extraction was attained via the addition of 1 M solution of sodium hydroxide. 

#### 3.4.3. Ultrasound-assisted Citric Acid Extraction

Betalains were extracted according to the method described by Singh et al. (2017) and Silva et al. (2018) [14,25] with some modifications. Freeze-dried beetroot powder (0.2 g, 230 µm) was extracted with 25 mL of citric acid solution (of varied pH), and continuous ultrasonication (Power 100 W; Frequency 42 kHz) at different time and temperature combinations as per the design given in Table 1. After ultrasound treatment, the mixture was centrifuged (SIGMA, Laborzentrifugation 3K10, Germany) twice at 9384× *g* for 30 min to obtain a clear supernatant. The extract was then stored at 4 °C until analysis.

### 3.5. Analysis of Betalains 

#### 3.5.1. Spectrophotometric Analysis of Total Betalains

Betalains were determined spectrophotometrically according to the method described in previous literature [19,47]. The sample extract (Section 2.4) was diluted 5 times before the spectrophotometer measurement (Cecil CE1011 Spectrophotometer, HACH, Manchester, UK) using McIlvaine buffer, which was prepared by mixing 30 mL of 0.1 M citric acid with 70 mL of 0.2 M sodium phosphate dibasic. The wavelengths used were 480 nm (for BX), 538 nm (for BC), and 600 nm (in order to account for any impurities). The measurement of BX and BC at 480 nm and 538 nm represents more than 95% of betalains present in beetroot sample. The expression used for the calculation of betalains is given by Equation (7) below.
(7)$$\mathrm{Betalains}\;(\mathrm{mg}\;\mathrm{of}\;\mathrm{BX}\;\mathrm{or}\;\mathrm{BC}/g\;\mathrm{of}\;\mathrm{dried}\;\mathrm{beetroot})=\frac{A\times \mathrm{DF}\times V\times \mathrm{MW}}{E\times L\times M}$$
where A = A_538_ − A_600_ for betacyanins (BCs) or A_485_ − A_600_ for betaxanthins (BXs); DF = dilution factor; MW (Molecular Weight) = 550 g/mol for betacyanin and 339 g/mol for betaxanthin; E = molar extinction co-efficient in Lmol^−1^ cm^−1^, and the values for betacyanins and betaxanthins are 60,000 and 48,000, respectively; V = volume of the extract; L= path length of quartz cuvette in cm and M= mass of dried sample taken for extraction.

#### 3.5.2. Identification and Quantification of Betacyanin and Betaxanthin by High Performance Liquid Chromatography (HPLC) to Validate Spectrophotometric Method

The analysis and quantification of betalains by HPLC was adapted from Nestora et al. (2016) [43] with some modifications in relation to the HPLC system, the detector and the column used for the separation of the analytes of interest. The detection of the betacyanins and the betaxanthins was carried out at 540 nm and 480 nm, respectively [41], and betanin (betacyanin) was used as the reference. The HPLC system (Agilent Technologies, Santa Clara, USA) consisted of diode array detector (DAD) with quadrupole solvent system. HPLC analyses were performed on a C18 reverse phase (RP) column (ZORBAX Eclipse XDB-C-18, Spectralab, Markham, Canada; 4.6 × 150 mm, 5 μm). The mobile phase consisted of 0.1% formic acid (eluent A), and HPLC grade acetonitrile (eluent B). The gradient program was as follows: 0 min 0% B, 13% B at 21 min, held at 13% B for 4 min, increased to 80% B at 30 min and held for 5 min. The flow rate was set to 1 mL/min, and the detection was monitored at 485 nm for BX and 538 nm for BC. The injection volume was 10 µL. Commercially available betanin containing a mixture of BC and BX was used to quantified BC and BX present in the extracts using an external calibration curve (concentration from 10 to 200 mg/mL; R^2^ = 0.99).

### 3.6. Total Phenolic Content 

Total phenolic content was estimated using previously reported methods [25,48,49] with some modifications. As a result of the lower temperature used, incubation time was increased by 30 min instead of the standard 40–45 min. A standard calibration curve of gallic acid (GA) was prepared using a stock solution of 1000 mg of GA/L (0.2 to 1.0 mg of GA/mL, R^2^ = 0.99). The procedure was as follows: 0.2 mL of the extracted sample was diluted with 2.8 mL of double distilled water, and 0.25 mL of Folin–Ciocalteau Reagent (FCR) was added. After 5 min of incubation, 0.75 mL of 20% Na_2_CO_3_ was added to the mixture and stirred using an auto stirrer for 30 s. After mixing the solution was stored for 90 min and measured at 760 nm using a spectrophotometer (Cecil CE1011 Spectrophotometer). The calculation for total phenolic content was carried out as per Equation (8) given below. The blank for the reference measurement was prepared with 0.2 mL of water instead of sample.
(8)$$\mathrm{Total}\;\mathrm{phenolic}\;\mathrm{content}\;(\mathrm{mg}\;\mathrm{of}\;\mathrm{GA}/g\;\mathrm{of}\;\mathrm{dried}\;\mathrm{beetroot})=\frac{C*V}{M}$$
where C = concentration of GA per mL of extract; V = volume of extract (mL); and M = amount of sample taken for extraction (g).

### 3.7. Predictive Modelling and Optimisation

The general method employed for the prediction and optimisation of the process parameters was RSM, and the predicted results of RSM were compared with ANN [38].

#### 3.7.1. Response Surface Methodology (RSM)

RSM was applied to develop the model, investigate the effect of process parameters and their interaction on the response variable which are yield of betalains and total phenolic content. The variables used for the optimisation of the amounts of betalains and total phenolics extracted are mentioned above. Design Expert (Version 11.0.0, Stat-Ease Inc., Minneapolis, MN, USA) was used to estimate the constants in the second-order polynomial given by Equation (9) below and draw the relevant surface response plots: (9)$$Y=\mathrm{Bo}+\sum_{i=1}^{k}B_{i}x_{i}+\sum_{i=1}^{k}B_{\mathrm{ii}}x_{i}x_{i}+\sum_{i=1}^{k-1}\sum_{j=i+1}^{k}B_{\mathrm{ij}}x_{i}x_{j}$$
where Y is the predicted response; Bo is the constant term; B_i_ is the linear coefficient; B_ii_ the squared coefficient; B_ij_ is the cross-product coefficient; i and j are the indices; $x_{i}\;\mathrm{and}\;x_{j}$ are the independent predictors and k is the number of factors.

#### 3.7.2. Artificial Neural Network

Feed forward architecture, where information flows layer wise in the forward direction, was used for predictive modelling by ANN [50]. The general structure of ANN model consists of three basic layers knowns as input layer, hidden layer, and output layer [38]. The independent variables were input parameters such as extraction time (Ut), extraction temperature (UT), and ethanol concentration (EC) in the case of extraction with ethanol; and Ut, UT and pH in the case of extraction with citric acid solution. The amounts of BC, BX, and TPC in the extract were the output parameters. ANN network has multiple internal parameters and one of these are weights, which is a real variable associated with two neurons in a network depending on the other parameters of the network, like number of iterations, and number of neurons [51]. The number of neurons in the input layer is simply the number of input or independent variables of the study, and it propagates the information to the hidden layer by scaling the input information via weights [39,52]. Consequently, the information received from the input layer into the hidden layer is processed in two steps. Firstly, the summation of the weighted input information of neurons that also sums bias as given by Equation (10) below [37].
(10)$$\mathrm{Sum}=\sum_{i=1}^{n}X_{i}W_{i}+b$$
where W_i_ is the weight function of the network, x_i_ is the input variables of the study, i denoting the indices, n is the number of input variables, and b is the bias of the network. The next step in the processing of hidden layer is to pass the weighted output through activation function whose role is to shift the space in non-linearity of input data [37]. The implementation of the logistic output function is given in Equation (11) below.
(11)$$f\left(\mathrm{sum}\right)=\frac{1}{1+\exp\left(-\mathrm{sum}\right)}$$

The output produced by the hidden layer becomes the input for the output layer, and the process to obtain the output from output layer is similar to the process of obtaining output from the hidden layer. To minimise the error between the experimental value and predicted, an error function is calculated as mean squared error (MSE). As training of an ANN model is an iterative process where these pre-defined model adequacies check error, and minimise it, by adjusting the weights and bias of the network appropriately. The formula for the calculation of error is given below in Equation (12).
(12)$$\mathrm{MSE}=\frac{1}{n}\sum_{i=1}^{n}{\left(y_{\exp}-y_{\mathrm{pred}}\right)}^{2}$$
where MSE is mean squared error; n is the number of total datasets; $y_{\exp}$ stands for experimental dataset used for making predictions; and $y_{\mathrm{pred}}$ represents the values predicted by the model. On the other hand, to check the deviation of the predicted values from the experimental dataset, root mean squared error (RMSE) and coefficient of determination (R^2^) were estimated for the entire dataset. R^2^ and RMSE are important parameters to establish the statistical deviation of the data across the central prediction line and reflect the accuracy of predictive modelling as given below in Equations (13) and (14).
(13)$$\mathrm{RMSE}=\sqrt{\frac{\sum_{1}^{n}\left(E_{\mathrm{pre}}-E_{\exp}\right)}{n}}$$
(14)$$R^{2}=1-\frac{\sum_{i=1}^{n}(E_{\mathrm{pre}}-E_{\exp})}{\sum_{i=1}^{n}(E_{m}-E_{\exp})}$$
where n is the number of experimental or predicted data; E_pre_ is the predicted value for each experimental results; E_exp_ is the experimental results; E_m_ is the average value of the observed experimental data; i denotes the index for each passing data. The training of the ANN model was carried out using a neural network feed-forward back-propagation algorithm, which is expected to take less time and memory for iterations [53]. The back-propagation training is based on the adjustments of two key network parameters, namely, the learning rate (0 < ɛ < 1), and momentum co-efficient (0 < γ < 1). The number of neurons in the hidden layer was optimised considering the lowest RMSE between the predicted and experimental values. The total number of neurons set for the optimisation was 1 to 60, and RMSE was measured for each increase in the number of neurons.

## 4. Conclusions

The extraction of betalains and the determination of total phenolic content from beetroot powder was performed using ultrasonication technology with conventional organic solvent of aqueous–ethanol and citric acid solution as solvents. Extraction using citric acid solution demonstrated a great potential for the extraction of polar compounds like betalains, and phenolics. Comparing the extraction efficiency of the both solvents, it could be concluded that the percentage yield using citric acid as a solvent in comparison to aqueous–ethanol was over 91% for BC, 92% for BX and 98% for TPC, respectively, which is sufficiently high to be considered as a potential solvent for the future extraction of such bioactive compounds. To optimise the extraction process for ethanol and citric acid solution as solvents a full factorial RSM design was implemented. The optimisation secured more than 90% of the betalains and 85% of the total phenolics in the extract for both solvent types. The resulting optimised conditions for citric acid solution as solvent was 10 min of extraction time at 30 °C of extraction temperature and pH 5, with yields of BC, BX, and TPC of 3.95 mg of BC/g, 3.54 mg BX/g, and 7.17 mg of GA/g of dried beetroot powder, respectively, with RSM desirability value of 0.928, whereas for aqueous–ethanol-optimised conditions, these were 15.8 min of extraction time at 20.1 °C of extraction temperature and 10% of ethanol concentration in water, with yields of BC, BX, and TPC as 4.15 mg of BC/g, 3.52 mg of BX/g, and 7.71 mg of GA/g of beetroot powder, respectively, with an RSM desirability value of 0.679. Therefore, the method developed can be successfully utilised for the efficient extraction of betalains and phenolics from beetroot to enable economic utilisation. The models developed using RSM and ANN were used to forecast future data and ANN proved to be a better predictive tool than RSM. 

In summary, the extraction of betalains and total phenolic compounds using citric acid as an alternative solvent approach opens a new possibility of performing extraction. In addition, it also opens other possibilities of exploring options available with ionic liquids (ILs) and natural deep eutectic solvents (NADES). Citric acid and other such food grade acids, which are commonly present in plant tissues, could be explored to develop NADES with the aim of optimising extraction procedures. 

## Figures and Tables

**Figure 1 molecules-28-06405-f001:**
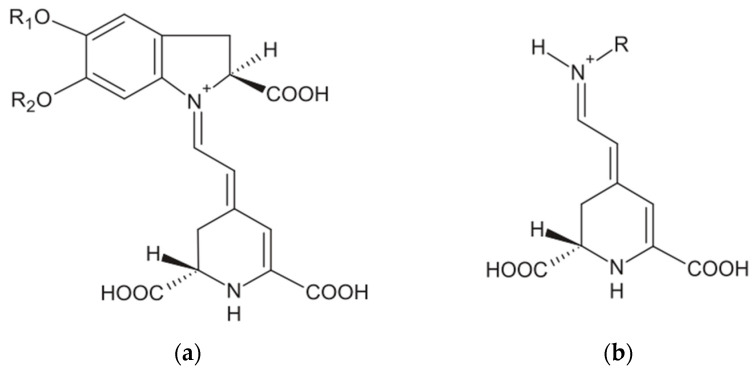
Chemical structure of betalains: (**a**) betacyanins and (**b**) betaxanthins.

**Figure 2 molecules-28-06405-f002:**
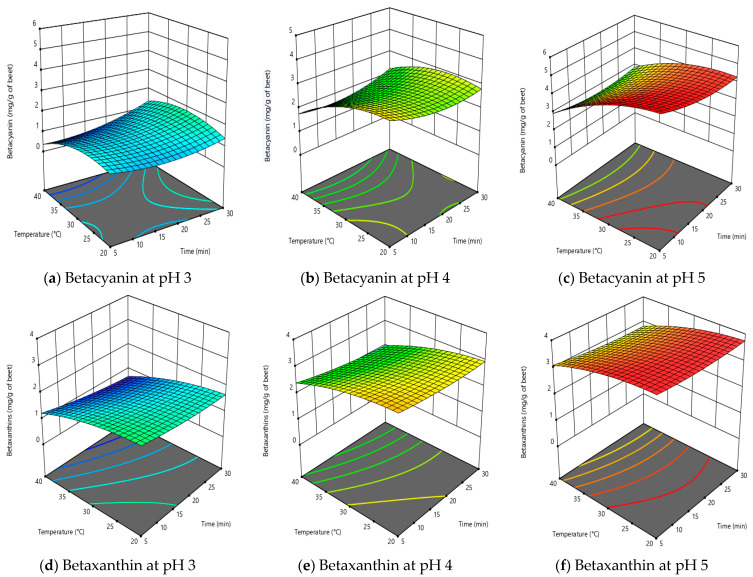
Response surface model plots for citric acid solutions as solvent; (**a**–**c**) show the effect of time and temperature on betacyanin at fixed pH; (**d**–**f**) show the effect of time and temperature on betaxanthin at fixed pH; and (**g**–**i**) show the effect of time and temperature on total phenolic content (TPC) at fixed pH.

**Figure 3 molecules-28-06405-f003:**
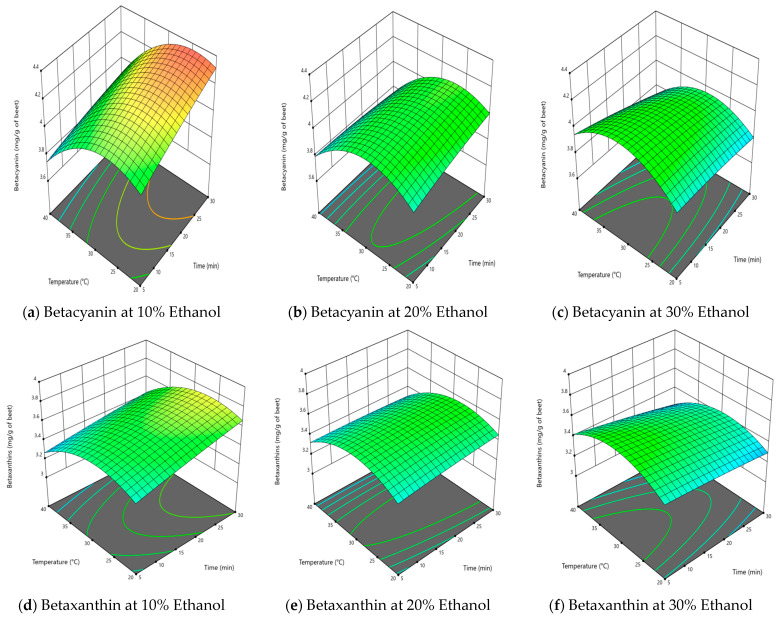
Response surface model plot where aqueous–ethanol was used as the solvent; (**a**–**c**) show the effect of time and temperature on betacyanin at fixed ethanol concentration; (**d**–**f**) show the effect of time and temperature on betaxanthin at fixed ethanol concentration; and (**g**–**i**) show the effect of time and temperature on total phenolic content (TPC) at fixed ethanol concentration.

**Figure 4 molecules-28-06405-f004:**
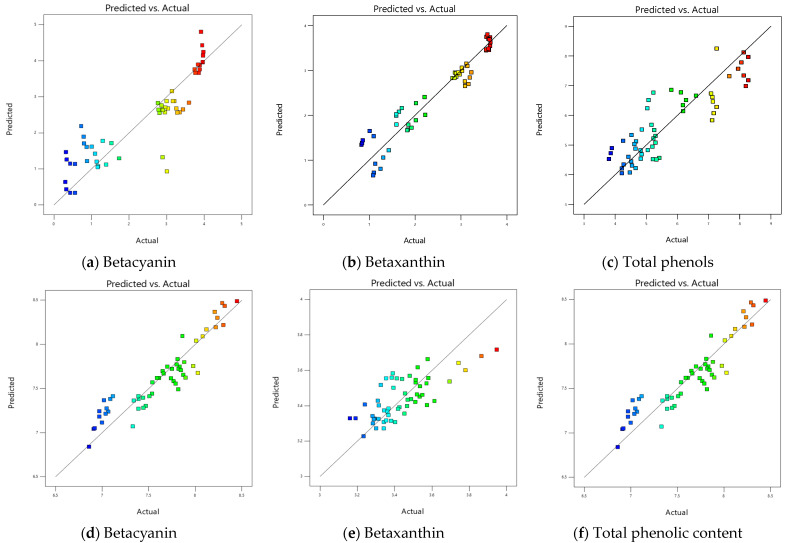
The correlation between predicted and actual values using Response Surface Methodology (RSM), (i) for citric acid as the solvent (**a**–**c**); (ii) ethanol as the solvent (**d**–**f**). The change in colour from blue to red indicates an increase in the concentration value.

**Figure 5 molecules-28-06405-f005:**
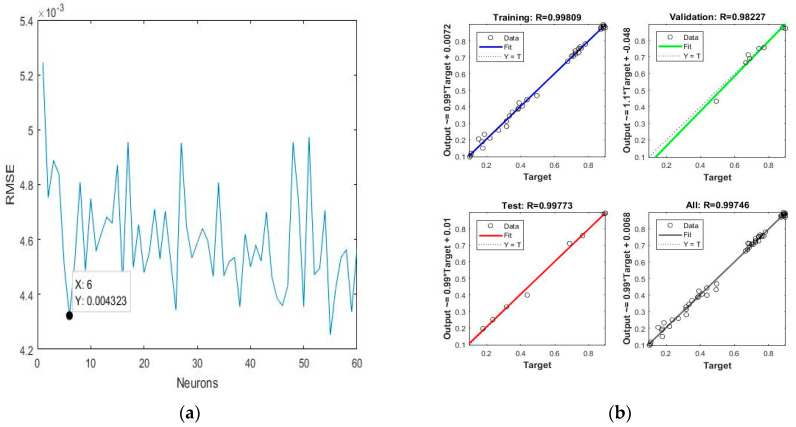
(**a**) Optimisation of the number of neurons against Root mean square error (RMSE); (**b**) regression between predicted and experimental data where citric acid solutions were used as the extraction solvent.

**Figure 6 molecules-28-06405-f006:**
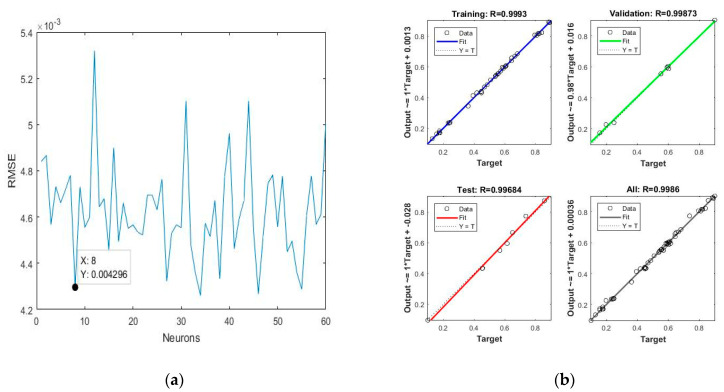
(**a**) Optimisation of the number of neurons against root mean square error (RMSE); (**b**) regression between predicted and experimental data where aqueous–ethanol was used as the extraction solvent.

**Figure 7 molecules-28-06405-f007:**
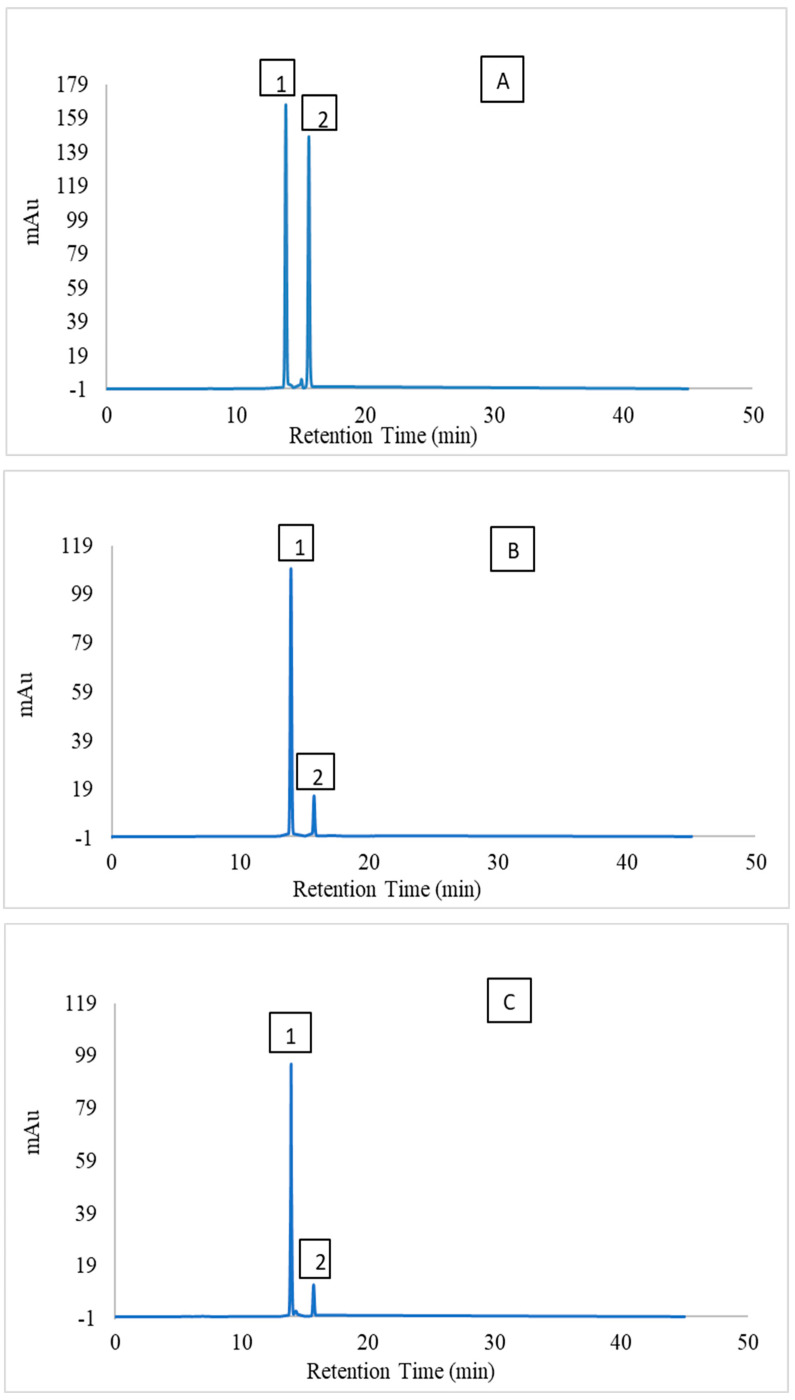
HPLC Chromatogram at 538 nm for betanin (1) and iso-betanin (2) (betacyanin and iso-betacyanin) in (**A**) Standard; (**B**) Ethanolic extract of optimised sample; (**C**) Citric acid extract of optimised sample.

**Table 1 molecules-28-06405-t001:** Factors applied for the ultrasonic extraction of betalains using aqueous–ethanol and citric acid solution as solvents.

Sl. No.	Variable Name	Variable Coding *	Range
1	Extraction Time (min)	Ut	5–30
2	Extraction Temperature (°C)	UT	20–40
3	Ethanol (%)	EC	10–30
1	Extraction Time (min)	Ut	5–30
2	Extraction Temperature (°C)	UT	20–40
3	Citric Acid Solution (pH)	pH	3–5

* Ut—Extraction Time (min); UT—Extraction Temperature (°C); EC—Ethanol (%); pH—Citric Acid Solution (pH).

**Table 2 molecules-28-06405-t002:** ANOVA table showing significance (*p*-values) of the different treatment parameters and solvents on betalains and total phenolic compounds.

	Solvent: Aqueous–Ethanol Solutions			Solvent: Citric Acid		
Parameters	Betacyanin	Betaxanthin	TPC	Betacyanin	Betaxanthin	TPC
A-Extraction Time (min) (Ut)	0.0233	---	0.0028	---	---	---
B-Extraction Temperature (°C) (UT)	<0.0001	0.0033	0.0043	<0.0001	<0.0001	<0.0001
Ethanol Concentration (%) (EC)/pH	0.0001	0.0364	<0.0001	<0.0001	<0.0001	<0.0001
AB	---	---	---	---	---	---
AC	0.0010	0.0034	---	---	---	---
BC	0.0001	---	<0.0001	0.0153	0.0153	0.0153
A^2^	---	---	0.0003	---	---	---
B^2^	<0.0001	<0.0001	0.0009	0.0222	0.0222	0.0222
C^2^	---	---	<0.0001	---	---	---
R^2^	0.887	0.799	0.863	0.782	0.892	0.794

AB—Extraction Time × Extraction Temperature. AC—Extraction Time × Ethanol Concentration or pH. BC—Extraction Temperature × Ethanol Concentration or pH. A^2^—Extraction Time × Extraction Time. B^2^—Extraction Temperature × Extraction Temperature. C^2^—Ethanol Concentration or pH × Ethanol Concentration or pH.

**Table 3 molecules-28-06405-t003:** Optimisation table and validation with real-time experiment for citric acid solution and aqueous–ethanol as solvent.

Sl. No.	Responses	Optimised Response	Average Real-Time experimental Value
1.	BC	3.95	3.91 ± 0.12
2.	BX	3.54	3.59 ± 0.23
3.	TPC	7.17	7.06 ± 0.36
Sl. No.	Responses	Optimised Response	Average Real-Time experimental Value
1.	BC	4.15	4.07 ± 0.15
2.	BX	3.52	3.68 ± 0.13
3.	TPC	7.71	7.65 ± 0.41

Responses are expressed as mg/g of beetroot powder for betacyanin (BC) and betaxanthin (BX), and as mg of gallic acid (GA)/g of beetroot powder for total phenolic content (TPC).

**Table 4 molecules-28-06405-t004:** Full factorial design matrix of independent variables and their corresponding experimental and predicted yields of total phenolic content betacyanin, and betaxanthin for citric acid solution as solvent.

				Experimental Responses			Predicted Responses by RSM			Predicted Responses by ANN		
Sl. No.	Time (min)	Temperature (°C)	pH	TPC (mg of GA/g)	BC (mg/g)	BX (mg/g)	TPC (mg of GA/g)	BC (mg/g)	BX (mg/g)	TPC (mg of GA/g)	BC (mg/g)	BX (mg/g)
1	5	40	5	5.26	2.44	2.42	5.51	2.88	3.13	5.31	2.39	2.41
2	10	40	5	5.30	2.34	2.33	5.32	2.67	2.99	5.34	2.43	2.34
3	15	40	5	5.30	2.28	2.29	5.08	2.56	2.91	5.25	2.35	2.28
4	20	40	5	4.82	2.17	2.23	4.82	2.55	2.86	4.76	2.20	2.23
5	25	40	5	4.83	2.15	2.20	4.53	2.64	2.83	4.74	2.11	2.20
6	30	40	5	4.66	2.13	2.17	4.22	2.83	2.83	4.74	2.11	2.17
7	5	40	4	5.17	1.18	1.70	5.68	1.71	2.4	5.23	1.08	1.64
8	10	40	4	4.86	0.78	1.56	5.52	1.61	2.27	4.84	0.80	1.45
9	15	40	4	4.53	0.67	1.32	5.33	1.61	2.16	4.48	0.79	1.36
10	20	40	4	4.67	0.62	1.27	5.12	1.71	2.08	4.69	0.71	1.30
11	25	40	4	4.65	0.61	1.23	4.87	1.89	2.02	4.66	0.59	1.26
12	30	40	4	4.43	0.56	1.22	4.62	2.18	1.99	4.40	0.58	1.22
13	5	40	3	5.20	0.44	1.10	4.94	0.33	1.22	5.11	0.48	1.07
14	10	40	3	5.05	0.33	1.01	4.83	0.33	1.05	4.93	0.31	1.05
15	15	40	3	4.85	0.25	0.87	4.68	0.43	0.92	4.88	0.26	0.97
16	20	40	3	5.33	0.24	0.96	4.51	0.63	0.80	5.21	0.23	0.94
17	25	40	3	4.54	2.32	0.85	4.31	0.93	0.72	4.65	2.34	0.90
18	30	40	3	4.47	2.23	0.84	4.07	1.32	0.66	4.35	2.02	0.88
19	5	30	3	5.23	0.68	1.23	5.22	1.21	1.79	5.14	0.69	1.13
20	10	30	3	4.26	0.44	0.78	5.14	1.13	1.65	4.42	0.50	0.82
21	15	30	3	4.62	0.33	0.85	5.03	1.14	1.53	4.43	0.39	0.77
22	20	30	3	3.90	0.26	0.66	4.89	1.26	1.44	4.08	0.37	0.68
23	25	30	3	3.86	0.25	0.65	4.72	1.47	1.38	4.06	0.26	0.65
24	30	30	3	3.80	1.00	0.64	4.52	1.77	1.34	3.85	0.89	0.63
25	5	30	4	7.08	2.77	2.49	6.73	2.84	2.95	6.99	2.70	2.48
26	10	30	4	7.12	2.65	2.46	6.61	2.65	2.84	7.09	2.56	2.42
27	15	30	4	7.14	2.53	2.38	6.46	2.56	2.75	7.03	2.41	2.40
28	20	30	4	7.26	2.58	2.44	6.28	2.57	2.69	7.15	2.46	2.42
29	25	30	4	7.17	2.51	2.38	6.07	2.67	2.66	7.30	2.46	2.40
30	30	30	4	7.12	2.47	2.38	5.83	2.88	2.65	7.23	2.47	2.34
31	5	30	5	8.13	3.35	2.81	7.34	4.24	3.63	8.13	3.20	2.75
32	10	30	5	8.28	3.33	2.80	7.18	3.95	3.54	8.17	3.20	2.79
33	15	30	5	8.20	3.30	2.77	6.98	3.76	3.48	7.96	3.22	2.80
34	20	30	5	5.23	3.28	2.78	6.76	3.66	3.45	5.35	3.36	2.78
35	25	30	5	5.08	3.20	2.73	6.51	3.66	3.44	5.11	3.36	2.75
36	30	30	5	5.03	3.07	2.73	6.23	3.76	3.46	4.97	3.18	2.72
37	5	20	3	5.42	1.34	1.71	4.56	1.29	2.01	5.56	1.42	1.63
38	10	20	3	5.21	1.07	1.55	4.52	1.12	1.89	5.37	1.28	1.57
39	15	20	3	4.52	0.90	1.43	4.44	1.05	1.79	4.56	1.05	1.51
40	20	20	3	4.28	0.91	1.48	4.34	1.07	1.72	4.26	0.88	1.47
41	25	20	3	4.22	0.88	1.42	4.21	1.23	1.68	4.30	0.84	1.43
42	30	20	3	4.22	0.85	1.41	4.05	1.42	1.66	4.16	0.80	1.41
43	5	20	4	5.81	2.42	2.40	6.86	3.15	3.15	5.72	2.39	2.43
44	10	20	4	6.12	2.31	2.32	6.77	2.88	3.06	6.01	2.37	2.37
45	15	20	4	6.60	2.28	2.33	6.66	2.75	2.99	6.44	2.34	2.32
46	20	20	4	6.28	2.23	2.27	6.51	2.63	2.95	6.22	2.30	2.28
47	25	20	4	6.18	2.22	2.23	6.34	2.65	2.94	6.13	2.24	2.29
48	30	20	4	6.19	2.21	2.21	6.13	2.76	2.95	6.19	2.21	2.30
49	5	20	5	7.26	3.31	2.75	8.25	4.80	3.8	7.23	3.37	2.74
50	10	20	5	8.13	3.37	2.80	8.12	4.42	3.74	7.99	3.34	2.73
51	15	20	5	8.28	3.36	2.77	7.96	4.14	3.7	8.42	3.32	2.74
52	20	20	5	8.05	3.36	2.79	7.78	3.96	3.69	8.16	3.32	2.78
53	25	20	5	7.95	3.29	2.77	7.56	3.88	3.75	7.73	3.37	2.78
54	30	20	5	7.67	3.29	2.74	7.32	3.89	3.75	7.59	3.38	2.75

RSM—Response Surface Methodology; ANN—Artificial Neural Network; TPC—Total Phenolic Compounds; BC—Betacyanin; BX—Betaxanthin; Responses are expressed as mg/g of beetroot powder for betacyanin (BC) and betaxanthin (BX), and as mg of gallic acid (GA)/g of beetroot powder for total phenolic compounds (TPC).

**Table 5 molecules-28-06405-t005:** Full factorial design matrix of independent variables and their corresponding experimental and predicted yields of total phenolic content betacyanin, and betaxanthin for aqueous ethanol as solvent.

				Experimental Data			Predicted Responses by RSM			Predicted Responses by ANN		
Sl. No.	Time (min)	Temperature (°C)	EC (%)	BC (mg/g)	BX (mg/g)	TPC (mg of GA/g)	BC (mg/g)	BX (mg/g)	TPC (mg of GA/g)	BC (mg/g)	BX (mg/g)	TPC (mg of GA/g)
1	5	40	30	4.04	3.61	7.92	3.95	3.43	8.09	3.96	3.57	7.93
2	10	40	30	3.81	3.43	8.16	3.91	3.39	8.30	3.91	3.51	8.15
3	15	40	30	3.98	3.45	8.31	3.87	3.35	8.43	3.88	3.43	8.30
4	20	40	30	3.81	3.38	8.32	3.83	3.31	8.49	3.84	3.37	8.33
5	25	40	30	3.77	3.30	8.30	3.78	3.27	8.47	3.79	3.31	8.31
6	30	40	30	3.72	3.23	8.25	3.73	3.23	8.37	3.68	3.26	8.27
7	5	40	20	3.65	3.16	7.85	3.80	3.33	7.83	3.68	3.17	7.88
8	10	40	20	3.69	3.19	8.02	3.80	3.33	8.04	3.72	3.20	8.09
9	15	40	20	3.77	3.29	8.18	3.79	3.33	8.17	3.74	3.24	8.17
10	20	40	20	3.71	3.29	8.21	3.78	3.32	8.22	3.74	3.29	8.18
11	25	40	20	3.76	3.35	8.17	3.77	3.32	8.19	3.77	3.34	8.17
12	30	40	20	3.79	3.40	8.13	3.75	3.31	8.09	3.79	3.42	8.15
13	5	40	10	3.88	3.34	7.34	3.75	3.27	7.07	3.88	3.41	7.31
14	10	40	10	3.89	3.34	7.39	3.78	3.31	7.27	3.90	3.35	7.43
15	15	40	10	3.82	3.28	7.44	3.81	3.34	7.40	3.91	3.33	7.46
16	20	40	10	3.89	3.34	7.51	3.84	3.37	7.44	3.91	3.33	7.47
17	25	40	10	3.86	3.31	7.50	3.86	3.46	7.41	3.87	3.32	7.47
18	30	40	10	3.86	3.31	7.49	3.88	3.43	7.3	3.81	3.31	7.46
19	5	30	30	4.03	3.39	7.56	4.10	3.58	7.28	4.04	3.40	7.56
20	10	30	30	4.03	3.39	7.76	4.08	3.56	7.49	4.06	3.38	7.75
21	15	30	30	4.17	3.51	7.87	4.05	3.53	7.62	4.11	3.38	7.89
22	20	30	30	4.04	3.39	7.89	4.03	3.5	7.68	4.13	3.41	7.90
23	25	30	30	4.12	3.46	7.85	4.00	3.47	7.66	4.12	3.45	7.84
24	30	30	30	4.14	3.47	7.80	3.96	3.43	7.56	4.10	3.50	7.76
25	5	30	20	4.12	3.57	7.49	4.03	3.53	7.41	4.14	3.63	7.48
26	10	30	20	4.20	3.69	7.70	4.05	3.54	7.62	4.09	3.60	7.69
27	15	30	20	4.06	3.51	7.78	4.06	3.55	7.75	4.02	3.54	7.78
28	20	30	20	3.97	3.44	7.79	4.07	3.55	7.87	3.98	3.47	7.79
29	25	30	20	3.94	3.41	7.77	4.07	3.55	7.77	3.98	3.40	7.78
30	30	30	20	3.88	3.35	7.74	4.07	3.55	7.67	3.99	3.35	7.77
31	5	30	10	3.95	3.54	6.96	4.07	3.51	7.04	3.98	3.53	6.97
32	10	30	10	4.00	3.58	7.03	4.12	3.56	7.24	4.07	3.60	7.09
33	15	30	10	4.21	3.78	7.12	4.17	3.62	7.37	4.16	3.67	7.11
34	20	30	10	4.16	3.74	7.13	4.22	3.64	7.41	4.24	3.76	7.11
35	25	30	10	4.29	3.87	7.11	4.26	3.68	7.38	4.29	3.87	7.10
36	30	30	10	4.38	3.95	7.10	4.29	3.72	7.27	4.35	3.95	7.10
37	5	20	10	3.84	3.24	7.53	4.01	3.41	7.38	3.98	3.31	7.53
38	10	20	10	4.22	3.52	7.71	4.08	3.46	7.58	4.14	3.44	7.68
39	15	20	10	3.99	3.33	7.78	4.14	3.52	7.71	4.07	3.43	7.75
40	20	20	10	4.22	3.48	7.78	4.26	3.57	7.75	4.16	3.48	7.78
41	25	20	10	4.26	3.52	7.76	4.26	3.62	7.72	4.24	3.54	7.76
42	30	20	10	4.32	3.58	7.72	4.31	3.66	7.62	4.31	3.62	7.72
43	5	20	20	3.96	3.42	7.49	3.88	3.38	7.36	4.01	3.46	7.48
44	10	20	20	4.08	3.57	7.60	3.91	3.44	7.57	4.06	3.50	7.62
45	15	20	20	4.05	3.51	7.67	3.94	3.42	7.74	4.08	3.53	7.67
46	20	20	20	4.02	3.49	7.69	3.97	3.44	7.75	4.07	3.55	7.69
47	25	20	20	4.05	3.53	7.68	3.99	3.45	7.72	4.06	3.56	7.69
48	30	20	20	4.06	3.55	7.66	4.01	3.46	7.62	4.05	3.55	7.66
49	5	20	30	3.88	3.47	6.86	3.85	3.4	6.84	3.88	3.43	6.85
50	10	20	30	3.75	3.37	6.91	3.85	3.38	7.05	3.82	3.41	6.92
51	15	20	30	3.76	3.36	6.96	3.85	3.37	7.18	3.78	3.38	6.99
52	20	20	30	3.78	3.37	6.99	3.84	3.35	7.24	3.75	3.35	6.99
53	25	20	30	3.74	3.31	7.00	3.82	3.32	7.21	3.73	3.33	6.99
54	30	20	30	3.73	3.28	6.99	3.81	3.34	7.12	3.72	3.30	7.01

**Table 6 molecules-28-06405-t006:** Statistical parameters to assess the predictive capability of the ANN for betacyanin (BC), betaxanthin (BX) and total phenolic content (TPC).

1. Citric Acid solution as Solvent				
Sl. No.	Responses	RMSE	MSE	R^2^
1	BC	0.032	0.092	0.99
2	BX	0.052	0.002	0.99
3	TPC	0.023	0.055	0.99
2. Aqueous–ethanol as solvent				
1	BC	0.051	0.003	0.99
2	BX	0.047	0.002	0.99
3	TPC	0.020	0.001	0.99

RMSE—root mean square error; MSE—mean squared error; R^2^—correlation coefficient.

**Table 7 molecules-28-06405-t007:** Statistical parameters to assess the predictive capability of the RSM models for betacyanin, betaxanthin and total phenolic content.

1. Citric acid solution as solvent				
Sl. No.	Responses	RMSE	MSE	R^2^
1	BC	0.600	0.101	0.78
2	BX	0.282	0.126	0.89
3	TPC	0.671	0.138	0.79
2. Aqueous–ethanol as solvent				
1	BC	0.471	0.211	0.88
2	BX	0.206	0.193	0.79
3	TPC	0.262	0.164	0.86

## Data Availability

All data provided in this manuscript.

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
