# Peer review of "A Comparative Study of Ethanol and Citric Acid Solutions for Extracting Betalains and Total Phenolic Content from Freeze-Dried Beetroot Powder"

_molecules, 2023, doi:10.3390/molecules28176405_

Round 1

Reviewer 1 Report

This research compares the extraction of betalains (betacyanin and betaxanthin) and total phenolic content using citric acid and ethanol/water solutions. 

After reading the manuscript, I have a few observations for improvement: 

Abstract and Introduction:

  1. The title is clear and adequately reflects the content of the research.
  2. The abstract provides a concise overview of the study, including the research objectives, methods, and main findings. However, it would be beneficial to include the rationale behind seeking an alternative solvent and the significance of finding a sustainable method.
  3. Specify the units for the extraction yields in the abstract (e.g., mg/g dry weight).
  4. In the abstract, briefly mention the specific extraction parameters that were investigated, such as temperature, time, solvent concentration, or any other relevant factors.
  5. The keywords are appropriate and cover the essential aspects of the study. However, consider adding "Betalain pigments" as a keyword to further enhance discoverability.
  6. The abstract could benefit from mentioning the key advantages of using citric acid as an alternative solvent over ethanol, as this is a crucial point of the study.
  7. Introduction: When citing references, please ensure that the format is consistent throughout the manuscript.
  8. In the sentence "Betalains consisting of two nitrogenous components, betacyanin (BC) and 41 betaxanthins (BX). These two nitrogeneous compounds can be of significant importance 42 to food, pharmaceuticals, cosmetics and dye industries, where it is also known as 'beetroot red' (Stintzing et al., 2003)," consider clarifying that betacyanin and betaxanthins are the two main classes of betalains found in beetroot. Additionally, it would be helpful to provide a brief explanation of their potential applications in the mentioned industries. Additionally, the paper cited are outdates. Please update to recent onces.
  9. 6. The sentence "Natural extracts from beetroot are used to replace synthetic colours in the food industry; in products such as confectionery and bakery, ice creams, yoghurts, and sweets (Azeredo, 2009)" could benefit from including specific examples of food products that use beetroot extracts as natural colorants. Also, the statement is outdated, please update.
  10. In the sentence "The stability of this colourant is pH and temperature dependent, and its application in high-temperature processed products is limited," provide some context or explanation as to why the stability is pH and temperature dependent and how this limitation affects its application in high-temperature processed products.
  11. It would be valuable to mention the specific environmental concerns associated with the use of organic solvents in betalain extraction, further highlighting the need for more sustainable extraction methods with updated references.
  12. In the sentence "Hence, there is a growing interest in the development of extraction procedures using alternative solvents which are perceived to be greener, cleaner, safer, and easier to adopt (Chemat et al., 2019)," consider elaborating on what makes citric acid a greener and safer alternative compared to traditional organic solvents.
  13. The sentence "The present study aims to overcome the limitations of earlier studies by investigating extraction in citric acid solution and aqueous ethanol solutions separately, and comparing the extracts obtained under otherwise identical conditions" is clear in its purpose. However, consider briefly mentioning the potential advantages of exploring these solvents separately and the significance of understanding their individual contributions.
  14. In the sentence "The use of ultrasound to intensify extraction has also been explored; and the independent and interactive effects of operating parameters such as the strength of the ethanol solution and pH of the citric acid solution, extraction time, extraction temperature, and ultrasound application have been evaluated using response surface methodology as well as artificial neural networks (ANN)," consider mentioning the benefits of using ultrasound in extraction and how it complements the study's objective.
  15. 12. The explanation of the ANN model is informative, but it could be helpful to mention how ANN outperforms or complements the response surface methodology (RSM) in predicting extraction yields.
  16. Objectives of the study is not clearly mentioned in the introduction. Please consider this in the last paragraph.

Materials

1.      In the "Sample Preparation" subsection (section 3.3), it is mentioned that the freeze-dried beetroot powder was ground and sieved, resulting in particles of average diameter 230 μm. If this particle size has been optimized for extraction, consider mentioning the reason for selecting this specific particle size and how it may impact the extraction efficiency.

2.      In the "Extraction of betalains" subsection (section 3.4), it is mentioned that the betalains were extracted using ethanol-water solvent and citric acid solution separately. Please specify the concentrations of ethanol used in each extraction, as it is mentioned that 10%, 20%, and 30% v/v ethanol solutions were employed in the experimental design.

3.      In the "Ultrasound assisted ethanolic extraction" and "Ultrasound assisted citric acid extraction" subsections (sections 3.4.1 and 3.4.3), include specific information about the ultrasonic parameters used, such as the frequency, power, and mode of operation (e.g., continuous or pulsed).

4.      Consider providing more information about the "modifications" made to the previously reported methods for the analysis of betalains and total phenolic content, as this will help readers understand the differences from the standard methods.

Results

1.      Line 83: stability of betalains is also likely to be reduced at the higher pH of water. Please elaborate why is this assumption being made?

2.      Very low pH is not favourable for industry as it will damage the equipment faster. Can the authors clarify why they include pH 3.0 in the optimization?

Author Response

Response to Reviewer 1 comments:

Reviewer 1: This research compares the extraction of betalains (betacyanin and betaxanthin) and total phenolic content using citric acid and ethanol/water solutions. After reading the manuscript, I have a few observations for improvement:

Abstract and Introduction:

Comment 1: The title is clear and adequately reflects the content of the research.

Response 1: Thank you for the positive comment.

Comment 2: The abstract provides a concise overview of the study, including the research objectives, methods, and main findings. However, it would be beneficial to include the rationale behind seeking an alternative solvent and the significance of finding a sustainable method.

Response 2: Thank you to the reviewer for the suggestion. This has now been addressed in the Abstract of the amended manuscript (Lines 11-15).

Comment 3: Specify the units for the extraction yields in the abstract (e.g., mg/g dry weight).

Response 3: Units for the extraction yields in the abstract have been added to the abstract (Lines 23-25).

Comment 4: In the abstract, briefly mention the specific extraction parameters that were investigated, such as temperature, time, solvent concentration, or any other relevant factors.

Response 4: This information has now been included in the abstract (Lines 17-19).

Comment 5: The keywords are appropriate and cover the essential aspects of the study. However, consider adding "Betalain pigments" as a keyword to further enhance discoverability.

Response 5: The word “Betalains” has been replaced with “Betalain pigments” as part of the keywords  (Line 33).

Comment 6: The abstract could benefit from mentioning the key advantages of using citric acid as an alternative solvent over ethanol, as this is a crucial point of the study.

Response 6: Authors have added the following information in the abstract (Lines 11-15).

“Using citric acid solution as a solvent offers several benefits over ethanol. Citric acid is a weak organic acid found naturally in citrus fruits, making it a safe and environmentally friendly choice for certain extraction processes. Moreover, the use of citric acid as solvents offers biodegradability, non-toxicity, non-flammability, and is cost effective”.

  1. Troter, D.; Zlatkovic, M.; Djokic-Stojanovic, D.; Konstantinovic, S.; Todorovic, Z. Citric Acid-Based Deep Eutectic Solvents: Physical Properties and Their Use as Cosolvents in Sulphuric Acid-Catalysed Ethanolysis of Oleic Acid. Adv. Technol. 2016, 5, 53–65, doi:10.5937/savteh1601053t.

Comment 7: Introduction: When citing references, please ensure that the format is consistent throughout the manuscript.

Response 7: This has been addressed and references are now consistent throughout the document.

Comment 8: In the sentence "Betalains consisting of two nitrogenous components, betacyanin (BC) and betaxanthins (BX). These two nitrogeneous compounds can be of significant importance to food, pharmaceuticals, cosmetics and dye industries, where it is also known as ‘beetroot red' (Stintzing et al., 2003)," consider clarifying that betacyanin and betaxanthin are the two main classes of betalains found in beetroot. Additionally, it would be helpful to provide a brief explanation of their potential applications in the mentioned industries. Additionally, the paper cited are outdates. Please update to recent ones.

Response 8:  Line 47 now reads “Betalains are classified into two different classes namely, betacyanins (BC) and betaxanthins (BX) (Figure 1)”. Furthermore, additional information has been added in relation to their potential applications in the mentioned industries (lines 50-59 in the amended manuscript).

References have also been updated:

  1. Luzardo-Ocampo, I.; Ramírez-Jiménez, A.K.; Yañez, J.; Mojica, L.; Luna-Vital, D.A. Technological Applications of Natural Colorants in Food Systems: A Review. Foods 2021, 10, 1–34, doi:10.3390/foods10030634.
  2. Khan, M.I. Plant Betalains: Safety, Antioxidant Activity, Clinical Efficacy, and Bioavailability. Compr. Rev. Food Sci. Food Saf. 2016, 15, 316–330, doi:10.1111/1541-4337.12185.

Comment 9: 6. The sentence "Natural extracts from beetroot are used to replace synthetic colours in the food industry; in products such as confectionery and bakery, ice creams, yoghurts, and sweets (Azeredo, 2009)" could benefit from including specific examples of food products that use beetroot extracts as natural colorants. Also, the statement is outdated, please update.

Response 9: Examples of food products that use beetroot extracts as natural colorants have been included and more up to date references have been included (Lines 50-59 in the amended manuscript).

Comment 10: In the sentence "The stability of this colourant is pH and temperature dependent, and its application in high-temperature processed products is limited," provide some context or explanation as to why the stability is pH and temperature dependent and how this limitation affects its application in high-temperature processed products.

Response 10: This has now been addressed in Lines 61-68 in the amended manuscript.

Comment 11: It would be valuable to mention the specific environmental concerns associated with the use of organic solvents in betalain extraction, further highlighting the need for more sustainable extraction methods with updated references.

Response 11: This information can now be seen in Lines 76-79 in the amended manuscript.

  1. Chemat, F.; Vian, M.A.; Ravi, H.K.; Khadhraoui, B.; Hilali, S.; Perino, S.; Tixier, A.S.F. Review of Alternative Solvents for Green Extraction of Food and Natural Products: Panorama, Principles, Applications and Prospects. Molecules 2019, 24, doi:10.3390/molecules24163007.

Comment 12: In the sentence "Hence, there is a growing interest in the development of extraction procedures using alternative solvents which are perceived to be greener, cleaner, safer, and easier to adopt (Chemat et al., 2019)," consider elaborating on what makes citric acid a greener and safer alternative compared to traditional organic solvents.

Response 12: This information has now been included in Lines 11-15 & 81-83 in the amended manuscript.

Comment 13: The sentence "The present study aims to overcome the limitations of earlier studies by investigating extraction in citric acid solution and aqueous ethanol solutions separately, and comparing the extracts obtained under otherwise identical conditions" is clear in its purpose. However, consider briefly mentioning the potential advantages of exploring these solvents separately and the significance of understanding their individual contributions.

Response 13: Lines 94-97 now reads: “This approach offers valuable insights into their solvent selectivity, yield of extraction, environmental impact, and process optimization. This knowledge is essential for advancing sustainable extraction practices and enhancing the utilization of betalains in various industries”.

Comment 14: In the sentence "The use of ultrasound to intensify extraction has also been explored; and the independent and interactive effects of operating parameters such as the strength of the ethanol solution and pH of the citric acid solution, extraction time, extraction temperature, and ultrasound application have been evaluated using response surface methodology as well as artificial neural networks (ANN)," consider mentioning the benefits of using ultrasound in extraction and how it complements the study's objective.

Response 14: Between lines 97-107, authors have established the utility of ultrasound extraction technology. Ultrasound is an intensive technology that required less solvent and accelerates the extraction process. This non-invasive method utilizes high-frequency sound waves to enhance the extraction process, making it more efficient and effective compared to traditional extraction methods. UAE offers several advantages, including reduced extraction times, lower energy consumption, and decreased reliance on organic solvents, making it a greener alternative.

Comment 15: The explanation of the ANN model is informative, but it could be helpful to mention how ANN outperforms or complements the response surface methodology (RSM) in predicting extraction yields.

 Response 15: Firstly, ANN and RSM performances were compared on the basis of parameters such as R2 , MSE and RMSE, and higher value of R accompanied by lower values of  MSE and RMSE indicate that ANN outperforms RSM. The comparison of experimental and predicted data for ANN (Figure 5) and RSM (Figure 4) also reinforced that ANN predictions are better than RSM. ANN has the ability to develop links between independent and dependent variables using weights and biases, whereas RSM uses equations upto second order level of interactions to form correlations (Line 354-356). This often results in ANN outperforming RSM in predicting responses. However, ANN can use the designs, data and responses developed by RSM to predict the new responses.  

Comment 16: Objectives of the study is not clearly mentioned in the introduction. Please consider this in the last paragraph.

Response 16:. Lines 91-97 now reads: “The present study aims to overcome the limitations of earlier studies by investigating ex-traction in citric acid solution and aqueous-ethanol solutions separately, and comparing the extracts obtained under otherwise identical conditions. This approach offers valuable insights into their solvent selectivity, yield of extraction, environmental impact, and pro-cess optimization. This knowledge is essential for advancing sustainable extraction practices and enhancing the utilization of betalains in various industries”.

Materials

Comment 1: In the "Sample Preparation" subsection (section 3.3), it is mentioned that the freeze-dried beetroot powder was ground and sieved, resulting in particles of average diameter 230 μm. If this particle size has been optimized for extraction, consider mentioning the reason for selecting this specific particle size and how it may impact the extraction efficiency.

Response 1: The specific particle size used in this study was selected based on our previous research -Ref 44. The following reference has now been added to the sub section 3.3 (Line 406)

  1. Kumar, R.; Oruna-Concha, M.J.; Methven, L.; Niranjan, K. Modelling Extraction Kinetics of Betalains from Freeze Dried Beetroot Powder into Aqueous Ethanol Solutions. J. Food Eng. 2023, 339, 111266, doi:10.1016/j.jfoodeng.2022.111266.

Comment 2: In the "Extraction of betalains" subsection (section 3.4), it is mentioned that the betalains were extracted using ethanol-water solvent and citric acid solution separately. Please specify the concentrations of ethanol used in each extraction, as it is mentioned that 10%, 20%, and 30%v/v ethanol solutions were employed in the experimental design.

Response 2:. Line 412 now specifies the concentrations of ethanol used in the study in accordance to the experimental design.

Comment 3: In the "Ultrasound assisted ethanolic extraction" and "Ultrasound assisted citric acid extraction" subsections (sections 3.4.1 and 3.4.3), include specific information about the ultrasonic parameters used, such as the frequency, power, and mode of operation (e.g., continuous or pulsed).

Response 3: Information about the ultrasonic parameters used, such as the frequency, power, and mode of operation information has now been added to (sections 3.4.1 and 3.4.3), Lines 412-413 and 424-425.

Comment 4: Consider providing more information about the "modifications" made to the previously reported methods for the analysis of betalains and total phenolic content, as this will help readers understand the differences from the standard methods.

Response 4: The details of modifications in the methods used to measure the betalains and total phenolic content have been provided in the amended manuscript in Lines 449-451 & Lines 465-466, respectively.

Results

Comment 1: Line 83: stability of betalains is also likely to be reduced at the higher pH of water. Please elaborate why is this assumption being made?

Response 1: Authors have mentioned that (Line 61) betalains are more stable in pH range of 4-5. Hence, it was fair to make this assumption.

Comment 2: Very low pH is not favourable for industry as it will damage the equipment faster. Can the authors clarify why they include pH 3.0 in the optimization?

Response 2: It is true that a low pH value must not be used in practice if it can be avoided. In this study, a pH value of 3 was only used as the lower limit of pH in the experimental design to assess its effect on extraction yields. The optimum value has been found to be higher anyway.

Reviewer 2 Report

In recent years, there has been a growing interest in the development of extraction procedures using alternative solvents which are perceived to be greener, cleaner, safer, and easier to adopt due to the use of organic solvents associated with environmental impact and safety concerns. Also, the industrial scale production, processing, packaging, retail market, and household consumption of beetroot more than 50% waste impact across the United Kingdom deemed it a research interest for mitigation. Hence, the extraction of betalains and the determination of total phenolic content from beetroot powder was performed using ultrasonication technology with conventional organic solvent of aqueous ethanol and citric acid solution, as solvents.

The research result merits the work to be considered for publication because the finding shows high accuracy in all samples tested per the utilized methodology and experimental design and will be of benefit to the food industry.

However, the following observations are recommended to merit the work for publication.

§  Line 10: dried plant material is vague be specific on the studied experimental sample which was “dried beetroot powder”

§  Line 9: “ethanol/water solution”, line: 16 “ethanol solution” and line 69: “aqueous ethanol” which of these was used?

§  Line 20: define RSM is it “response surface methodology” in line 74

§  Line 71-77: need to be revised for clarity. Ultrasound use in the extraction process must be well established as the study aims to consider the operation parameters studied. ANN, MSE, RMSE, and R2 must be clearly stated as part of the study aim.

§  Table 1: define variable coding UT, Ut, and EC below the table for clarity

§  Line 97: “Extraction was not a significant factor of betalain …when citric was used as solvent”. However, subsequent lines discuss the results of extraction time. Correct statement for clarity else it seems contradictory.

§  LINE 107: “GA” or “GA/g” check and correct accordingly

§  Line 155: due “to”

§  Check space in line 86, 85, 112, 113, 116, 125, 162, 153, 175 BX, 180, 183, 190, 193, 194, 233, 259, 297, 340, 341,342, 348, 349, 381, 396, 418, 437, 554, 557 and “table 1 B-extraction”

§  Line 186: “Roriz et al., (2017)” or “Roriz et al. (2017)” check and correct accordingly

§  Figure 3: image (a) looks blurry, improve the image resolution

§  Table 5: “MSE” check font size

§  2.7 HPLC analysis: revised the discussed section with optimization functionalities that make the identified chromatogram align with the standard. Also, highlight on its mechanism revealing the identified chromatogram corroboration with results in Table 3.

§  Line 430-431: italicize the section 3.5.2 Identification and quantification of betacyanin and betaxanthin by high performance liquid chromatography (HPLC) to validate spectrophotometric method

§  Line 459: “Total Phenolic content” or “Total Phenolic Content” check and correct accordingly

§  Line 479: equation 9, any definition of variables Xi, Xj, i, j check and correct accordingly

§  Line 501: “xi” check font size and also any definition of variables n and j check and correct accordingly

§  Line 557-562: revise the section for clarity. The statement is too long and confusing.

Author Response

Response to Reviewer 2 comments:

Reviewer #2:  In recent years, there has been a growing interest in the development of extraction procedures using alternative solvents which are perceived to be greener, cleaner, safer, and easier to adopt due to the use of organic solvents associated with environmental impact and safety concerns. Also, the industrial scale production, processing, packaging, retail market, and household consumption of beetroot more than 50% waste impact across the United Kingdom deemed it a research interest for mitigation. Hence, the extraction of betalains and the determination of total phenolic content from beetroot powder was performed using ultrasonication technology with conventional organic solvent of aqueous ethanol and citric acid solution, as solvents.

Comment 1: Line 10: dried plant material is vague be specific on the studied experimental sample which was “dried beetroot powder”.

Response 1: Line 11 in the amended manuscript now reads: “The aim is to find an environmentally sustainable alternative solvent for extracting these compounds from dried beetroot powder”.

Comment 2: Line 9: “ethanol/water solution”, line: 16 “ethanol solution” and line 69: “aqueous ethanol “which of these was used?

Response 2: The word “aqueous-ethanol” has now been used throughout the document

Comment 3: Line 20 (now line 30): define RSM is it “response surface methodology” in line 74

Response 3: Lines 15 now reads: “A full factorial design and response surface methodology (RSM) were employed to assess the effects of extraction parameters”.

Comment 4: Line 71-77: need to be revised for clarity. Ultrasound use in the extraction process must be well established as the study aims to consider the operation parameters studied. ANN, MSE,RMSE, and R must be clearly stated as part of the study aim.

Response 4: Authors have addressed this comment and tried establishing a solid foundation for the application of ultrasound in this study Lines 97-114. They have also highlighted how the statistical and model parameters helped in designing a better predictive model using RSM and ANN.

Comment 5: Table 1: define variable coding UT, Ut, and EC below the table for clarity

Response 5: The definition for all variable coding has been added to Table 1 (Lines 130-131).

Comment 6: Line 97: “Extraction was not a significant factor of betalain …when citric was used as solvent”. However, subsequent lines discuss the results of extraction time. Correct statement for clarity else it seems contradictory.

Response 6: Statement in Line 97, now Line 137 in the amended manuscript is in line with the results obtained in the present study when using citric acid as solvent. Further explanation was provided subsequently, outside the timing used in the present study and based in previous published research, which could have incurred a significant effect.

Comment 7: LINE 107: “GA” or “GA/g” check and correct accordingly.

Response 7: GA/g has now been corrected in lines 146 and 155 in the amended manuscript.

Comment 8: Line 155: due “to”

Response 8: due “to the” has now been corrected in Line 191 in the amended manuscript.

Comment 9: Check space in line 86, 85, 112, 113, 116, 125, 162, 153, 175 BX, 180, 183, 190, 193,194, 233, 259, 297, 340, 341,342, 348, 349, 381, 396, 418, 437, 554, 557 and “table 1 B-extraction”.

Response 9: A uniform line spacing of 1.5 has been implemented throughout the document.

Comment 10: Line 186: “Roriz et al., (2017)” or “Roriz et al. (2017)” check and correct accordingly

Response 10: Line 218 in the amended manuscript now reads “Roriz et al., (2017).

Comment 11: Figure 3: image (a) looks blurry, improve the image resolution

Response 11: Figure 3 has now become Figure 4, because of earlier amendments (addition of an extra Figure 1: Chemical structure of Betalains).  The image resolution for Figure 4(a) has now been improved.

Comment 12: Table 5: “MSE” check font size

Response 12: The font for “MSE” in Table 5 has been corrected.

Comment 13: 2.7 HPLC analysis: revised the discussed section with optimization functionalities that make the identified chromatogram align with the standard. Also, highlight on its mechanism revealing the identified chromatogram corroboration with results in Table 3.

Response 13: The compounds of interest (betalains) were identified against the standard of betanin (betacyanin) based on retention times and UV spectra in the optimized samples for both aqueous-ethanol and citric acid extracts.

Comment 14: Line 430-431: italicize the section “3.5.2 Identification and quantification of betacyanin and betaxanthin by high performance liquid chromatography (HPLC) to validate spectrophotometric method”

Response 14: Line 430-431 now Line 447-48 is now italicized.

Comment 15: Line 459: “Total Phenolic content” or “Total Phenolic Content” check and correct accordingly

Response 15: “Total Phenolic Content” has now been used throughout the document.

Comment 16: Line 479: equation 9, any definition of variables Xi, Xj, i, j check and correct accordingly

Response 16: The definition of variables Xi, Xj, i, j is now described in Lines 493-494 (previously Line 479).

Comment 17: Line 501: “xi” check font size and also any definition of variables n and j check and correct accordingly

Response 17: Authors have checked font size and also defined the used variables. (Lines 515-516).

Comment 18: Line 557-562: revise the section for clarity. The statement is too long and confusing.

Response 18: The statement in Lines 557-562, now Lines 577-582 in the amended manuscript has been made clearer and it now reads:

“In summary, extraction of betalains and total phenolic compounds using citric acid as an alternative solvent approach opens a new possibility of performing extraction. In addition, it also opens other possibilities of exploring options available with ionic liquids (ILs) and natural deep eutectic solvents (NADES). Citric acid and other such food grade acids, which are commonly present in plant tissues, could be explored to develop NADES with the aim of optimising extraction procedures.”

Reviewer 3 Report

This study aims to find an environmentally sustainable alternative solvent for the extraction of betalains and total phenolic content from freeze-dried beetroot powder. This paper is interesting and optimized extraction  conditions well, but the following points should be considered.

1. Please give the structural formula for betalains.

2. In Figure 3, an explanation should be added in the caption as to what each color symbol indicates.

3. A, B and C in Figure 5 should use the same type in the text.

4. One line space should be left above the title of Table 3.

5. Conclusions are not supported by the results. Please show the extraction rates of each compound with aqueous ethanol and citric acid solutions under optimum conditions in table.

Author Response

Response to Reviewer 3 comments

This study aims to find an environmentally sustainable alternative solvent for the extraction of betalains and total phenolic content from freeze-dried beetroot powder. This paper is interesting and optimized extraction conditions well, but the following points should be considered.

Comment 1: Please give the structural formula for betalains.

Response 1: The chemical structure for the two main classes of betalains is now represented in Figure 1.

Comment 2: In Figure 3, an explanation should be added in the caption as to what each color symbol indicates.

Response 2: Figure 3 is now Figure 4 in the amended manuscript. Authors have clarified what the colour symbol represents by adding the following information in the caption of Figure 4: “The change in colour from blue to red indicates an increase in the concentration value”.

Comment 3: A, B and C in Figure 5 should use the same type in the text.

Response 3: Figure 5 is now Figure 6 in the amended manuscript. This has been corrected and the same nomenclature for Figure 6 (namely A, B and C) is used in both the figure caption and in the text (Lines 362-363).

Comment 4: One line space should be left above the title of Table 3.

Response 4: This has now been corrected (Line 264).

Comment 5: Conclusions are not supported by the results. Please show the extraction rates of each compound with aqueous ethanol and citric acid solutions under optimum conditions in table.

Response 5: Details of extraction rates for BC, BX and TPC with aqueous ethanol and citric acid solutions at optimum conditions (as indicated in Table 3) have now been incorporated in the conclusion section (Lines 566-573).